# Pre-Processing to Increase the Capacity of SAG Mill Circuits—Case Study

Homero Delboni, Jr. [1], Evandro Costa e Silva [2,*], Vladmir Kronemberger Alves [3] and Ana Carolina Chieregati [1]

[1] Department of Mining and Petroleum Engineering, University of São Paulo, São Paulo 05508-070, Brazil; hdelboni@usp.br (H.D.J.); ana.chieregati@usp.br (A.C.C.)
[2] Vale S.A.—Exploration and Mineral Projects, Rio de Janeiro 22250-145, Brazil
[3] Department of Mining Engineering, Federal University of Ouro Preto, Ouro Preto 35400-000, Brazil; vladmir.alves@ufop.edu.br
[*] Correspondence: evandro.costaesilva@gmail.com; Tel.: +55-31-99801-4417

**Abstract:** This paper describes the adopted approach for increasing the capacity of an existing industrial grinding circuit by adapting the respective configuration to process the ore from a new mine. Accordingly, due to Sossego mine exhaustion, Vale S. A. decided to use the existing industrial facilities and infrastructure for processing the Cristalino ore deposit located in Para state, within the Brazilian Amazon. Considering the higher hardness of Cristalino ore compared to Sossego ore, a reduction in capacity in the existing SAG grinding circuit was anticipated. A comprehensive grinding pilot plant campaign was conducted with a characterization program including 98 Cristalino ore samples, as described throughout this paper. Sossego grinding circuit was also surveyed for mathematical modeling and simulations to assess such an estimative further. The mathematical model calibration for setting different circuit configurations and operating conditions to enhance the circuit's capacity was based on the combination of pilot plant results and ore characterization. Simulations indicated that a capacity increase of 12% would be achieved in the existing grinding circuit by further crushing 35% of SAG mill fresh feed. Such figures would represent yearly additions of 8.3 kt in copper and 250 kg in gold productions.

**Keywords:** grinding; SAG; pre-processing; copper ore

## 1. Introduction

A steady increase in the world copper demand is predicted for the next 25 years of, essentially due to global economic recovery for both developing and developed countries. Therefore, mining companies are currently concentrating their efforts on geological exploration, developing new mines, and expanding existing copper ore operations worldwide.

Carajás is a prolific mineral province located in Pará state, within the Amazon region in Brazil, containing not only world-class iron ore deposits, but also significant gold, manganese, and copper deposits, the latter occurring as IOCG-Iron Oxide Copper Gold geological formations. The Carajás IOCG deposits show primary and secondary copper sulphides, together with native and oxidized copper minerals, in combination with gold and magnetite. Cristalino is one of those IOCG deposits with reserves of 200 Mt @ 0.71% Cu and 0.3 g/t Au, together with 25 Mt of high-grade hematite iron ore occurring om top of the copper mineralization.

Despite the significant reserves, the capital costs related to infrastructure, logistics, and tailing disposal reduce the economic feasibility of most copper deposits in the Carajás region. Therefore, instead of considering each deposit as an individual unit, Vale is evaluating the use of existing industrial facilities to process the ores from those individual deposits, currently considered satellites. The exhaustion of the Sossego mine is due to occur in 2024 and according to such an approach, the Sossego mine industrial installations will be available to process the Cristalino ore deposit. Additional positive aspects of such

an arrangement are the relatively short distance between the Cristalino deposit and the Sossego industrial plant, as well as using Sossego's mined pits for future tailing disposal for minimizing costs and reducing environmental impacts.

Cristalino certified reserves include 200 Mt of copper and gold ores, averaging 0.71% Cu and 0.3 g/t Au and a substantial volume (~25 Mt) of high-grade hematite cap over the main copper ore body. Due to the existing iron ore production and logistic system in the region (Carajás and S11D), Vale decided to install a 6 Mtpa dry plant to process such an iron ore deposit. It comprises of a three-stage crushing and screening, whose product is predicted to be 90% passing size ($P_{90}$) of 19 mm. As the estimated LOM (Life of Mine) for the iron operationprocessing is 4–6 years, the industrial crushing installation will become available to process other ores or be relocated to another operation.

### 1.1. Objective

The present work aims to assess additional capacity scenarios for processing the Cristalino ore in the Sossego industrial grinding circuit by adjusting the SAG mill fresh feed size distribution, as resulting from the crushing and screening industrial installation.

### 1.2. Literature Review

Semi-Autogenous grinding (SAG) is a long-proven and mature technology used in the last 60 years in several industrial comminution circuits to process various ore types, according to ever-increasing capacities [1]). Several different circuit configurations are adopted in industrial installations, including open or closed circuits with a classifier, stand-alone mode (single-stage), or coupled with a ball mill in SAB configuration [2]. Full secondary pre-crushing of the fresh mill feed is an effective alternative to increasing the SAG mill throughput [3].

The high capacity provided by a single crushing-milling line and associated handling equipment simplifications are the two main aspects that significantly reduce the capital costs of SAG industrial plants. The ability to process a wide range of ore types, including sticky and clayey ones, is also a popular aspect of SAG milling [4].

Ore hardness and size distribution are key factors that dictate the required grinding energy for providing the stipulated product size and throughput for a given installation. These two factors, combined with mill dimensions and operating conditions, modulate the charge dynamics within the SAG mill chamber and, therefore, the process performance.

Coarse and competent ores require relatively high breakage energy which is provided by bigger SAG mills and installed power. In such conditions abrasion is the driving breakage mechanism for coarse particles, while the attritioning prevails for breaking relatively small particles. The balance between these two breakage mechanisms controls the tendency to generate intermediate size material, or critical size fraction within the SAG mill chamber. Such a critical shows low breakage kinetics as it is not small enough to be broken by attritioning, therefore buiding up within the mill chamber. On the other hand, fine and friable ores are generally processed with high ball charges in SAG mills to compensate for the absence of coarse particles as grinding media.

As the feed size distribution commands the charge volume and thus the SAG mill power draw associated with competent ores, various industrial operations later included additional crushing stages for increasing the throughput of existing SAG mill circuits [5]. In the case of such modification, the SAG mill product is coarser compared with the corresponding relatively coarse feed size distribution, thus resulting in a higher energy requirement for ball milling in SAB–Semi-autogenous ball mill circuits. Balancing operating conditions between primary (SAG milling) and secondary (ball milling) stages are thus the key factor for maximizing the grinding circuit capacity [5–7].

*1.3. Sossego Grinding Circuit*

In 2004, Vale started the Sossego mine in Canaã dos Carajás, Pará state, in the Brazilian Amazon. The industrial plant was designed to process 15 Mt/year (41 kt/day) of a copper-gold ore (0.98% Cu and 0.28 g/t Au).

The SABC, an acronym for Semi-Autogenous Ball mill pebble Crusher configuration, was adopted for the Sossego industrial grinding circuit under a nominal throughput of 1841 t/h for a product with an 80% passing size ($P_{80}$) of 0.21 mm [8]. Accordingly, a long-distance conveyor belt conveys the primary crusher product to a primary stockpile. From the stockpile, the ore is reclaimed and transferred to the 11.6 m diameter (38′) by 7.0 m length (19′) single SAG mill equipped with a 20 MW electric motor. The SAG mill product flows to two 3.6 m (12′) by 7.3 m (24′) horizontal screens, whose combined oversize (pebbles) is diverted to two Metso MP 800 cone crushers. The crushed product returns to the SAG mill feed, closing the primary grinding circuit. The horizontal screen undersize is pumped to two separate secondary grinding lines reversely configured. Each grinding line comprises a single 0.84 m (33″) cyclone nest, together with a 6.7 m diameter (22′) by 9.7 m length (29′) single ball mill equipped with an 8.5 MW electric motor. The combined cyclone nest overflow is the grinding circuit product [9,10]. Figure 1 illustrates the Sossego comminution circuit flow sheet.

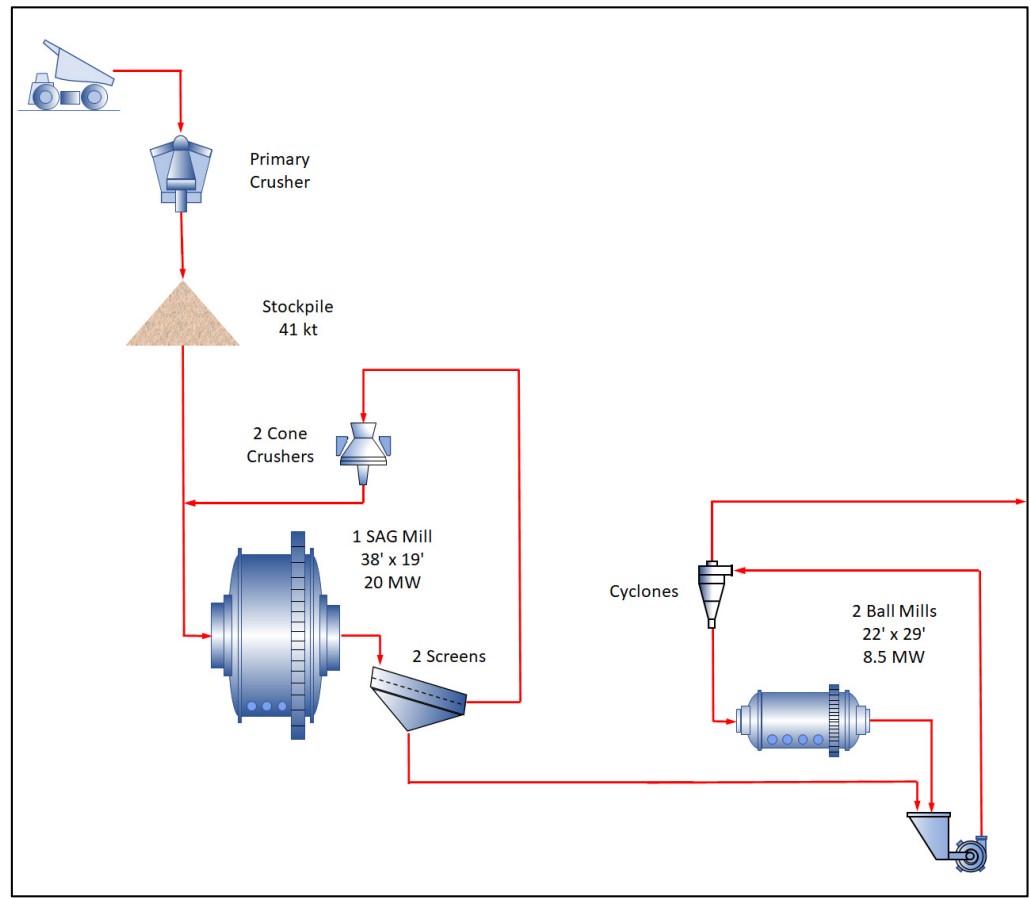

**Figure 1.** Flow sheet of the Sossego comminution circuit.

## 2. Experimental

*2.1. Ore Characterization-Bench Scale Test Work*

A comprehensive testing campaign was conducted for Cristalino selected samples. The sample characterization program consisted of determining comminution properties for all 98 samples, including Drop Weight Tests (DWT), Abrasion testing, and the Bond ball mill work index (BWi), the latter determined by using a 0.21 mm closing screen.

In the Drop Weight device, individual particles placed on a steel anvil are impacted by a dropping weight under gravity [4]. The resulting size distribution from broken fragments is reduced in terms of the *tn* values, where *tn* is defined as the percentage passing each *n*th fraction of the original particle size. The selected value for representing the breakage is $t_{10}$, which can be thus described as the percentage passing in one-tenth of the original particle size. The $t_{10}$ is related to the specific comminution energy as follows:

$$t_{10} = A\,(1 - e^{-b\,\text{Ecs}}) \tag{1}$$

Where Ecs is specific comminution energy (kWh/t), $t_{10}$ is the percentage passing 1/10th of the initial mean particle size tested, and *A* and *b* are ore impact breakage parameters. The product of parameters A and b are referred as Breakage Index is regarded as an index of the ore's amenability to breakage by impact [11]. An alternative Drop Weight test procedure was used in cases of sample size and quantity limitations [12].

### 2.2. Pilot Plant Campaign

A comprehensive Cristalino pilot plant grinding campaign was carried out at *Centro de Investigaciones Mineras y Metalurgicas*–CIMM facilities in Santiago, Chile. A total of 120 t of Cristalino ore was prepared and sent to CIMM, where it was crushed and screened prior to grinding tests.

The processing equipment included a 1.83 m (8′) diameter by 0.61 m (2′) length AG/SAG mill equipped with a 20 kW motor, a cone crusher, a 0.91 m (3′) diameter by 1.22 m (4′) length ball mill equipped with a 15 kW motor, together with two spiral classifiers [13]. Figure 2 shows both AG/SAG and ball mill circuit configurations adopted in the pilot campaign.

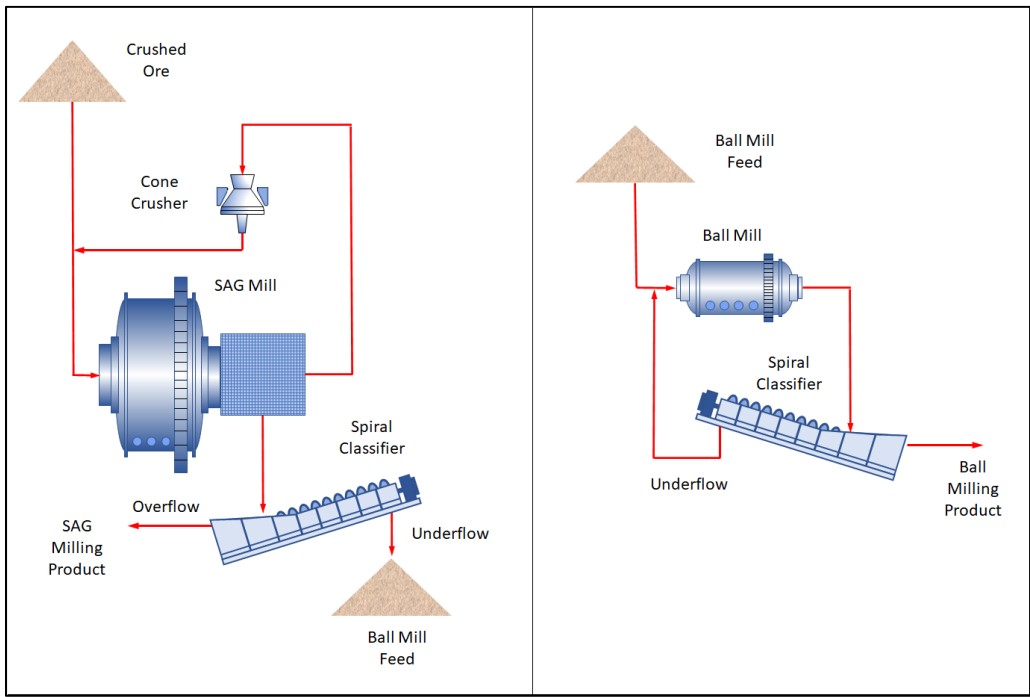

**Figure 2.** Circuit configurations adopted in the pilot plant campaign.

The AG/SAG milling stage included an optional pebble crushing and a spiral classifier to separate the final product (overflow) from the ball mill circuit feed (underflow). Accordingly, the spiral classifier underflow from the AG/SAG circuit was later homogenized and processed at the secondary grinding circuit configured in a direct mode with a ball mill and a spiral classifier.

The SAG mill–pebble crushing circuit closed configuration is referred to as SAC, while the opencircuit configuration is SAC-A. Twelve AG/SAG tests were conducted according to different configurations and operating conditions. Two additional ball mill tests were also carried out with coarse products obtained from SAC and SAC-A tests. Figure 3 shows the pilot SAG mill, trommel, and spiral classifier.

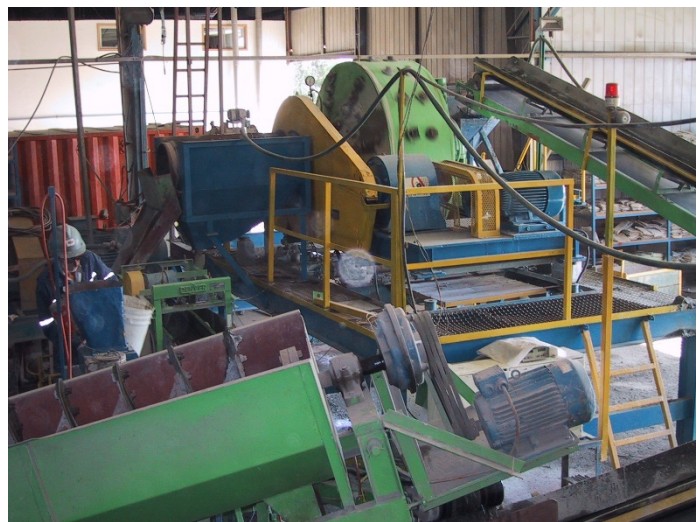

**Figure 3.** SAG mill, trommel, and spiral classifier.

## 3. Results and Discussion

### 3.1. Ore Characterization

Table 1 summarizes the results obtained from characterization tests on 98 selected Cristalino samples. Accordingly, the listed Breakage Index results averaged 31.1 for the 98 samples, which indicates a high tenacity to single-particle fragmentation. The high tenacity result also corroborates for adopting SAG milling for Cristalino ore, as it should provide resilient grinding media in such mills.

**Table 1.** Summary of characterization test results.

| Index | Wi (kWh/t) | Breakage Index | S.G. |
|---|---|---|---|
| Mean | 17.4 | 31.1 | 3.13 |
| Standard Deviation-SD | 1.5 | 5.1 | 0.22 |
| Coefficient of Variation (%) | 9 | 16 | 7 |
| Mean + One SD | 18.9 | 36.2 | 3.35 |
| Mean − One SD | 15.9 | 26.0 | 2.91 |
| Maximum | 19.8 | 51.1 | 3.85 |
| Minimum | 13.4 | 22.3 | 2.77 |
| Lower Quartile | 16.1 | 27.9 | 2.99 |
| Median | 17.6 | 30.2 | 3.11 |
| Upper Quartile | 19.8 | 51.1 | 3.85 |

BWi results listed in Table 1 show an average of 17.4 kWh/t, thus indicating a relatively high resistance to grinding in ball mills, together with a very small coefficient of variation (9%), which is also an indicator of a highly homogeneous distribution. If SABC is the adopted configuration for processing Cristalino ore, power distribution among SAG mill and ball milling should consider such an aspect.

Homogeneity was also observed in the specific gravity-SG distribution indices listed in Table 1, here emphasized by the respective low coefficient of variation (7%). The relatively low upper quartile SG value indicated the low risk associated with the preferential accumulation of high SG material in the SAG mill charge, which is regarded as deleterious

to SAG milling due to increased power draw, as well as difficulties in maintaining a steady-state operation.

### 3.2. Pilot Plant Campaign

The AG/SAG pilot plant campaign with Cristalino ore consisted of 12 tests, as shown in Table 2. Eight tests were configured as SAC, three tests according to SAC-A configuration, as well as one single fully autogenous test in a closed-circuit configuration (FAC) [13].

**Table 2.** Cristalino AG/SAG milling pilot plant campaign results.

| Test | Circuit Configuration | Feed Size Distribution | Mill Speed (% of Critical) | Ball Charge (%) | Pebble Ports Open | Fresh Feed Flow Rate (kg/h) | Mill Power Draw-Net (kW) | Specific Energy (kWh/t) | Circuit Product $P_{80}$ (mm) |
|------|----------------------|------------------------|----------------------------|-----------------|-------------------|------------------------------|---------------------------|--------------------------|-------------------------------|
| SAG-1 | SAC (*) | Sossego | 72 | 10 | 4 | 1242 | 10.7 | 8.6 | 5.6 |
| SAG-2 | SAC | Cristalino | 72 | 10 | 4 | 1366 | 10.7 | 7.8 | 6.6 |
| SAG-3 | SAC-A (**) | Cristalino | 72 | 10 | 4 | 1504 | 10.9 | 7.2 | 3.4 |
| SAG-4 | FAC (***) | Cristalino | 72 | 0 | 4 | 667 | 7.5 | 11.3 | 5.0 |
| SAG-5 | SAC-A | Cristalino | 72 | 12 | 4 | 1561 | 11.5 | 7.4 | 4.2 |
| SAG-6 | SAC | Cristalino | 75 | 10 | 4 | 1443 | 10.6 | 7.4 | 7.0 |
| SAG-7 | SAC-A | Cristalino | 75 | 10 | 4 | 1521 | 11.1 | 7.3 | 4.7 |
| SAG-8 | SAC | Secondary Crushing | 72 | 10 | 4 | 1664 | 11.0 | 6.6 | 7.0 |
| SAG-9 | SAC | Coarse | 75 | 10 | 4 | 1350 | 10.9 | 8.1 | 6.7 |
| SAG-10 | SAC | Cristalino | 72 | 10 | 4 | 1343 | 10.9 | 8.1 | 6.0 |
| SAG-11 | SAC | Cristalino | 75 | 12 | 4 | 1403 | 11.4 | 8.1 | 7.1 |
| SAG-12 | SAC | Cristalino | 75 | 12 | 2 | 1270 | 11.7 | 9.2 | 6.2 |

(*) SAC: semi-autogenous mill with pebble crusher; (**) SAC-A: semi-autogenous mill with pebble crusher in open mode; (***) FAC: fully autogenous test in a closed-circuit configuration.

Three feed size distributions were tested throughout the pilot plant campaign. The one referred to as "Cristalino" was derived from blasting and primary crushing simulations, while the second ("Sossego") was based on data obtained from sampling the Sossego industrial operation. The third SAG mill fresh feed, referred as "Secondary Crushing", simulated the size distribution generated from a crushing circuit configured with primary and secondary stages. In all but one test, there were four open pebble ports, which consisted of relatively large apertures (76 mm × 76 mm) located around the mill grate. Table 2 also describes the test results in terms of the specific energy consumption, i.e., the ratio between mill power draw and fresh feed flow rate, together with the $P_{80}$ of the trommel undersize (circuit product).

Tests SAG-1, SAG-2, SAG-8, and SAG-9 show the influence of fresh feed size distribution on circuit performance. In this case, a coarser distribution as per Sossego ($P_{80}$ = 123 mm) resulted in greater specific energy consumption (8.6 kWh/t) compared to the Cristalino size distribution ($P_{80}$ = 112 mm), the latter indicating 7.8 kWh/t. Interestingly, the combination between the coarsest feed size distribution ($P_{80}$ = 141 mm) and higher mill speed as per the SAG-9 test (75% of critical) resulted in intermediate specific energy consumption (8.1 kWh/t). Such a result was interpreted as SAG-2 providing a relatively adequate amount of media for grinding. In contrast, the excessive coarse material in SAG-9 resulted in a burden to mill charge even with a higher mill speed. Conversely, the relatively fine size distribution in SAG-1 indicated an absence of grinding media provided by the coarse material.

Additional crushing provided by a secondary crusher enhanced the grinding circuit performance by reducing the corresponding specific energy to 6.6 kWh/t, as observed in test SAG-8. The latter consisted of the smallest specific energy value observed for the entire pilot plant campaign. Such a figure represents a significant 15% reduction in energy consumption compared to the SAG-2 test result, the latter indicating 7.8 kWh/t.

Tests SAG-2 and SAG-3 show the performance associated with circuit configuration. Greater grinding specific energy was obtained for the pebble crusher closed configuration (7.8 kWh/t) than the corresponding open one (7.2 kWh/t). Even though the difference in specific grinding energy was relatively small, the latter figure should include the additional energy associated with ball milling with a coarser feed. Opening the SAG mill circuit

is thus an option for balancing the power distribution between primary and secondary grinding stages in the case of adopting the SABC circuit configuration for processing the Cristalino ore. The specific energy value (7.4 kWh/t) obtained of both SAG-5 and SAG-6 tests were interpreted as equivalent to an increased ball charge and mill rotating speed. Both changes were thus beneficial to the SAG mill performance compared with the SAG-2 test performance (7.8 kWh/t), where both ball charge and mill speed were relatively low. Such a result is typical of high tenacity ores that are better ground by increased breakage energy. The latter is provided by a greater ball charge (higher overall steel grinding media mass) or higher mill speed (higher impact height).

Based on SAG-7 test results, the lack of circulating load in such a scenario showed virtually no effect on the SAG circuit performance. Conversely, the combined effect between excessive impact provided by greater ball charge (12%) and increased mill speed (75%) was deleterious to the mill performance, as shown in the SAG-11 test.

Reduction in the number of open pebble ports was even more detrimental to mill performance, as shown in the SAG-12 test, compared to the SAG-11 test. These two tests clearly show the influence of critical size accumulation in the SAG mill charge even with relatively high ball charge and mill speed. In such a context, the SAG-4 test also corroborates the necessity of providing a high-energy environment in the mill chamber for breaking the critical size material. The lack of steel media in autogenous grinding showed the highest specific energy consumption (11.3 kWh/t) in the entire testing campaign.

The preferential accumulation of critical size material in the SAG mill charge is illustrated in Figure 4, showing the SAG-2 test ore charge after screening. The relatively high volume of the 2″ (51 mm) size fraction is evident, referred to as critical size material.

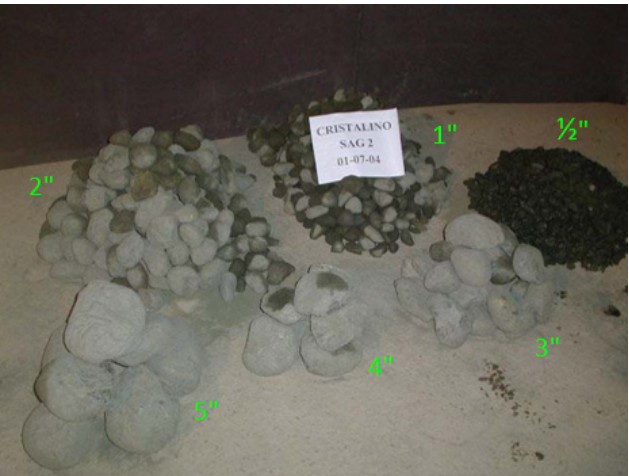

**Figure 4.** SAG mill internal ore charge after SAG-2 test

The accumulation of critical size material in the SAG mill charge is thus reduced by a specific combination between mill speed and ball charge (SAG-5 and SAG-6 tests), eliminating the circulating load (SAG-3 test) or, better of them all, including a secondary crushing stage to significantly reduce the amount of critical size material in the SAG mill fresh feed [14].

## 4. Simulations

Processing the Cristalino ore on Sossego industrial comminution circuit was simulated to assess the performances associated with each selected scenario, including combinations between crushing and grinding stages. Section 4.1 describes the simulations carried out for obtaining different SAG mill fresh feed size distributions, while Section 4.2 shows the results obtained in the Sossego grinding circuit.

### 4.1. Crushing Circuit

Cristalino's crushing circuit was simulated according to different combinations between primary crushing and two supplementary crushing stages, as per the flowsheet shown in Figure 5.

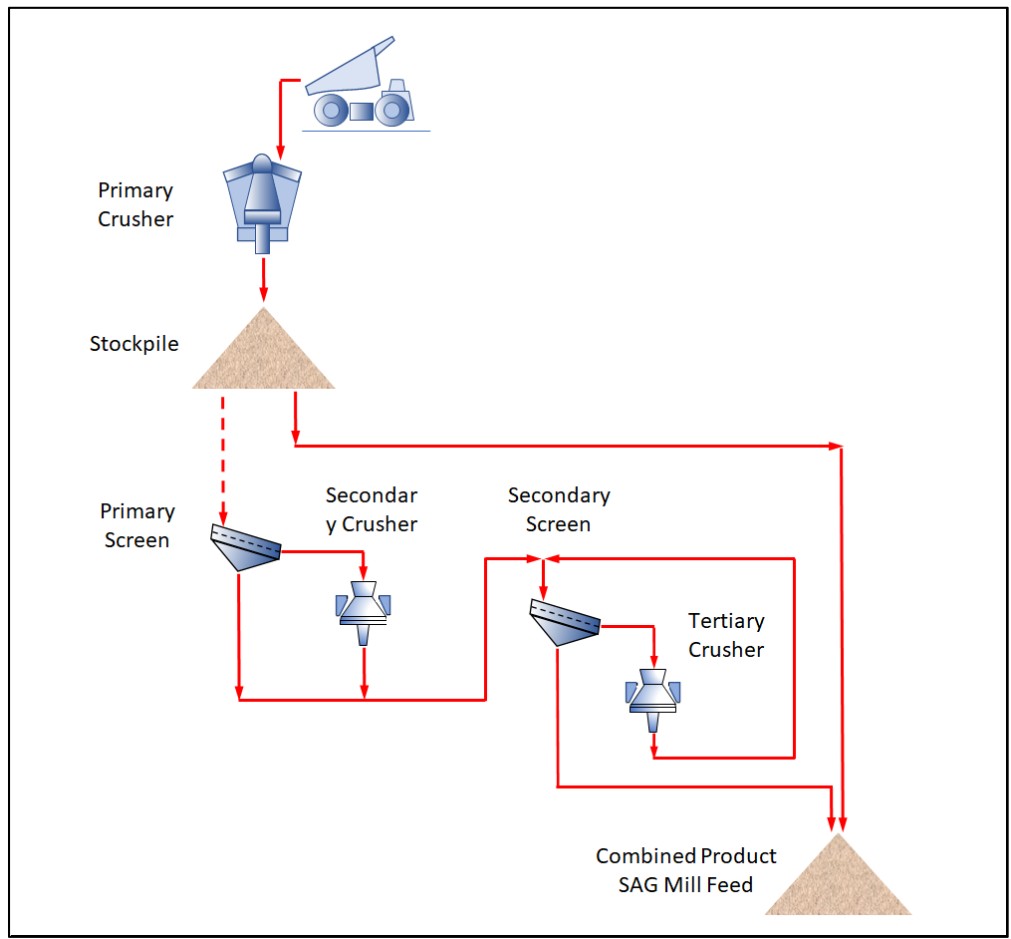

**Figure 5.** Cristalino crushing circuit.

According to Figure 5, mining hauling trucks dump the run of mine directly onto a gyratory crusher, whose product is conveyed to a primary stockpile. The reclaimed ore from the primary stockpile is split between two products according to adjusted ratios. The first product is diverted directly to the secondary stockpile, while the second product is conveyed to the supplementary crushing plant, whose product is also transferred to the same secondary stockpile. In this case, the secondary crushing stage included a double deck vibrating screen (primary) designed to operate in an open and reverse configuration with a cone crusher (secondary). Both upper and lower deck oversize fractions are diverted to the cone crusher feed, whose product is conveyed, together with screen undersize, to the tertiary crushing stage.

Based on the flowsheet shown in Figure 5, the tertiary crushing stage included a double deck vibrating screen (secondary) designed to operate in closed and reverse configuration with a cone crusher (tertiary). Similarly, both upper and lower deck oversize fractions are diverted to the cone crusher feed, whose product is conveyed back to the screen feed, therefore closing the circuit operation. The secondary screen undersize is the final secondary/tertiary plant product, which is conveyed to the secondary stockpile.

In summary, the simulated Cristalino crushing plant product comprises a blended size distribution between primary and secondary/tertiary stages, as shown in Figure 5.

The primary crusher simulation was based on the blasting pattern adopted for Cristalino, together with the stipulated operation for the selected gyratory crusher. The resulting primary crusher product size distribution is listed in Table 3, which indicates a 300 mm top size and 110 mm $P_{80}$.

**Table 3.** Cristalino crushing circuit size distributions.

| Size (mm) | Primary Crushing Product (Coarse) | Tertiary Crushing Product (Fine) | Blended Crushing Circuit Product * | | |
|---|---|---|---|---|---|
| | | | **Accumulated Percent Passing** | | |
| | | | 40% Fines | 35% Fines | 25% Fines |
| 300 | 100 | 100 | 100 | 100 | 100 |
| 203 | 94.5 | 100 | 96.7 | 96.4 | 95.9 |
| 127 | 84.1 | 100 | 90.5 | 89.7 | 88.1 |
| 102 | 77.8 | 100 | 86.7 | 85.6 | 83.4 |
| 76.2 | 67.0 | 100 | 80.2 | 78.6 | 75.3 |
| 50.8 | 48.8 | 100 | 69.3 | 66.7 | 61.6 |
| 25.4 | 30.8 | 99.8 | 58.4 | 55.0 | 48.1 |
| 19.1 | 18.4 | 95.4 | 49.2 | 45.3 | 37.6 |
| 12.7 | 12.0 | 69.7 | 35.1 | 32.2 | 26.4 |
| 6.35 | 8.2 | 39.4 | 20.7 | 19.1 | 16.0 |
| 3.36 | 5.8 | 22.6 | 12.5 | 11.7 | 10.0 |
| 1.00 | 3.3 | 8.9 | 5.5 | 5.2 | 4.7 |
| $P_{80}$ (mm) | 110 | 14 | 76 | 81 | 90 |

(*) 40% Fines; 35% Fines; 25% Fines—percentage of primary crusher discharge crushed processed on the secondary + tertiary crushing circuit.

Table 3 also indicates the size distribution derived from simulations carried out based on a typical secondary/tertiary crushing circuit, resulting in a 26 mm top size and a $P_{80}$ of 14 mm. Both size distributions were considered typical based on industrial comminution circuits.

Table 3 also shows the calculated size distributions derived from blending the two size distributions, i.e., primary crusher product, referred to as "coarse", and tertiary crusher product, in this case, referred to as "fine". The three blended products listed in Table 3 were calculated according to 40%, 35%, and 25% of fines, in this case referring to the amount (%) of tertiary crusher product in the corresponding blend. The $P_{80}$ were 76 mm, 81 mm, and 90 mm, respectively.

### 4.2. Grinding Circuit

The blended products from crushing circuit listed in Table 3 were used to simulate the processing of Cristalino ore on the Sossego grinding circuit, including both primary (SAG milling) and secondary (ball milling) stages. Simulations were carried out on the JKSimMet simulator based on previously calibrated mathematical models as obtained from a thorough survey campaign on the Sossego industrial grinding plant, followed by fitting the respective mathematical model related to each piece of processing equipment [10].

Preliminary simulations were conducted in the primary grinding stage to assess the effects of the fresh feed size distribution in the SAG mill charge. In these cases, the main adopted criterion was to obtain a constant charge volume comprising coarse ore fragments, steel grinding media and fines. The reference charge volume was 28%, resulting from the above-referred survey carried out in the Sossego SAG mill. Even though such a value may be regarded as relatively high, it was considered robust for a mature industrial grinding operation like Sossego.

The selected primary grinding stage scenarios were combined with secondary stage simulations. In this case, the adopted criterion was to maintain a constant ball mill charge volume, thus resulting in the same power draw figures as obtained in the Sossego survey. The targeted grinding circuit product $P_{80}$ was 0.210 mm.

Integrated simulation scenarios indicated that increasingly finer fresh feed size distribution progressively reduced the total charge volume in the SAG mill, which in turn enabled significant increases in primary circuit throughput. However, the secondary grinding stage limited such higher throughput scenarios, basically due to limitations in the ball mill power draw.

All three blended size distributions, previously listed in Table 3 were simulated in the integrated Sossego grinding circuit to assess the balance between primary and secondary stages for processing the Cristalino ore. Various additional aspects were evaluated in each scenario, such as circulating loads, pump flow rates, number of cyclones in operation. The selected simulated alternative is listed in Table 4, which comprises of the 65% coarse and 35% fine blend. Such an alternative was compared with the primary crushing only scenario, listed in Table 4 as Base Case.

**Table 4.** Summary of grinding circuit simulations.

| Variable | | Base Case | Simulation |
|---|---|---|---|
| **Grinding Circuit Fresh Feed** | | **Primary Crusher Product** | **Blended-Coarse (65%) and Fines (35%)** |
| Circuit Throughput-Solids | t/h | 1570 | 1758 |
| | Increase (%) | - | 12 |
| Circulating Load-SAG Milling | t/h | 295 | 259 |
| | % | 19 | 15 |
| Circulating Load-Ball Milling | t/h | 5126 | 6520 |
| | % | 326 | 371 |
| Slurry Pulp Pump Flow Rate | m$^3$/h | 6026 | 7449 |
| | Increase (%) | - | 24 |
| Cyclones in Operation | - | 7 | 9 |
| Cyclone Underflow Dilution | % | 80.7 | 80.7 |
| Total Charge Volume-SAG mill | % | 30.0 | 27.3 |
| Power Draw-SAG Mill | kW | 14,696 | 14,542 |
| Power Draw-Ball Mills (each) | kW | 8211 | 8211 |
| Specific Energy-SAG | kWh/t | 9.4 | 8.3 |
| Specific Energy-Ball Mills | kWh/t | 10.5 | 9.3 |
| Specific Energy-SAG and Ball Mills | kWh/t | 19.8 | 17.6 |
| | Reduction (%) | - | 11 |
| Grinding Circuit P$_{80}$ | mm | 0.210 | 0.210 |

Table 4 also grinding circuit throughputs of 1570 t/h and 1758 t/h, respectively for the Base Case and the blended feed scenario, thus representing a significant 12% increase in circuit capacity. In this case, the overall circuit throughput was clearly limited by the secondary stage capacity, as the SAG mill total load was smaller than the targeted 28% figure.

Due to the finer (blended) SAG mill fresh feed, Table 4 shows a smaller circulating load in the primary circuit, i.e., 15%, compared to 19% in the Base Case. The opposite though was observed in the secondary grinding stage, where a higher circulating load (371%) was obtained for the blended feed, as compared with Base Case (326%).According to the same scenario, the pumping flow rate was also significantly increased by 24% in the blended feed scenario, as compared with the corresponding Base Case figure. Such figures emphasize the secondary circuit limitations in terms of milling power. The resulting overall specific energy was 11% smaller for the blended feed scenario (17.6 kWh/t) than the Base Case (19.8 kWh/t).

## 5. Conclusions

A comprehensive pilot plant grinding campaign with Cristalino ore was carried out, including different circuit configurations, feed size distribution, mill speed, and steel

ball charge. The results indicated a particularly significant specific energy reduction with additional crushing, as compared with primary crushing only. These results were combined with those obtained from a characterization testing program in simulation exercises aiming to assess different scenarios for processing the Cristalino ore in the existing Sossego industrial grinding plant.

The following conclusions were derived from the work here described:

1.  Breakage Index results averaged 31.1 for the 98 tested samples, thus indicating a high tenacity for single-particle fragmentation. The calculated coefficient of variation indicated a relatively low value (16%), showing a highly homogeneous distribution regarding the single-particle fragmentation for tested Cristalino samples;

2.  The Bond Wi averaged 17.4 kWh/t for the same 98 Cristalino samples, thus indicating a relatively high resistance to grinding in ball mills, which is an indicator of potential high energy consumption in the secondary grinding stage of a typical SABC circuit configuration;

3.  Pilot plant testing on Cristalino ore resulted in 7.8 kWh/t specific energy consumption for the reference SAC configuration (SAG-2). Further tests indicated that: (a) a reduction to 7.4 kWh/t by increasing the mill speed (SAG-6), (b) an increased to 9.2 kWh/t by reducing the number of pebble ports open (SAG-12), (c) an increase to 8.1 kWh/t by coarsening the feed size distribution together with increasing the mill speed (SAG-9), as well as (d) a reduction to 6.6 kWh/t by further crushing the feed size distribution (SAG-8);

4.  The 15% specific energy reduction obtained by including additional crushing in the SAG mill feed (SAG 8 test), as compared with refrence test (SAG-2) was the basis for simulating the processing of Cristalino ore at the Sossego industrial grinding circuit;

5.  Simulations of a hybrid crushing circuit for Cristalino ore indicated significant different size distributions of primary crushing product (coarse) and secondary/tertiary crushing products (fine);

6.  Simulations of Sossego grinding circuit processing Cristalino ore indicated that increasingly finer fresh feed size distribution significantly reduced the SAG mill load, therefore increasing the SAG mill throughput. However, the secondary grinding stage throughput limited such higher throughput scenarios essentially due to limitations in the ball mill installed power;

7.  Simulations of Sossego grinding circuit showed that a blended feed comprising of 35% fines and 65% coarse resulted in 12% increase in the overall grinding circuit throughput, as compared with coarse feed only scenario;

8.  The calculated energy consumption for entire grinding circuit (SAG and ball milling) was reduced by11% for the selected blend, as as compared with coarse feed only scenario;

9.  Simulations indicated potential increase in throughput by installing additiaonal ball milling in the industrial circuit.

**Author Contributions:** Conceptualization, E.C.e.S.; methodology, H.D.J.; validation, V.K.A. and A.C.C.; formal analysis, H.D.J., V.K.A. and A.C.C.; investigation, H.D.J. and E.C.e.S.; data curation, H.D.J.; witing-original draft, E.C.e.S.; writing-review and editing, H.D.J. and A.C.C.; supervision, E.C.e.S.; project administration, E.C.e.S. All authors have read and agreed to the published version of the manuscript.

**Funding:** All funding for the described testwork was provided by Vale S.A.

**Data Availability Statement:** Not applicable.

**Acknowledgments:** The authors wish to thank Vale S.A. for sponsoring the work and giving permission to publish it.

**Conflicts of Interest:** Authors declare no conflict of interest.

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
