# Peer review of "Pre-Processing to Increase the Capacity of SAG Mill Circuits—Case Study"

_minerals, doi:10.3390/min12060727_

Round 1
Reviewer 1 Report
The paper concerns increasing the efficiency of the existing industrial grinding system by adjusting its appropriate configuration to the mechanical processing of ore from a new mine.
This issue is extremely important in the context of the emerging raw materials and energy problems in the world. The shortage of primary raw materials is becoming more and more noticeable, their price is increasing, which makes the use of these raw materials in traditional manufacturing technologies less and less profitable. The growing energy demand, resulting from the growing number of people as well as the development of civilization, forces the need to look for energy savings in production processes. There was a need to replace the linear method of producing goods with production in a closed cycle. This necessity applies to both the raw material aspect and the energy aspect.
In this context, the paper takes on special importance. The authors took up the problem of analyzing the use of the existing inferior sources of raw materials, as well as reducing the energy consumption of their processing.
The undoubted advantage of the paper is the use of the results of experimental research carried out on an industrial scale in simulation processes. This allows to expect that the obtained results of mathematical simulations will be reliable.
The paper has significant shortcomings:
1. The presented results (Table 2 - Delboni Jr., 2004; Figure 4.- Paredes et al., 2004.) seem unclear in terms of originality. It would be better to emphasize what is, in this paper, the original aspect of using the results presented in the cited papers.
2. Mathematical simulation is an essential part of the paper. In this context, it seems that the model (Mendonça et al., 2015) and the conditions and results of its calibration are described too modestly.
3. The use of the terms comminution, grinding, milling would require re-analysis.
Are the identical title names (points 2.1 and 3.1; 2.2. and 3.2.) correct or are there any editorial errors?
Author Response
Please see the attachement. It contains further corrections, as pointed by the reviwer, particularly the following responses to each one reviwer's comments:
- The phrase "These are original results, as obtained specifically for such a campaign" was included in the text to emphasize such an aspect
- Specific model calibration constants were considered intelectual property by Vale. They are thus protected and not allowed to be published.
- Titles were carefully reviwed and corrected accordingly

Reviewer 2 Report
This paper is useful to investigate the SAG mill property.
Please correct or add the following points to be more understandable.
Abstract
Cristalino ore deposit ( ). Please write the country.
Introduction
P90 ( ). Please write 90% passing product size?
Also, 1.3, P80( )
Please write the reference of Cristalino and Sossego.
1.3 Please write the abbreviation words. SABC ( )?
p.4 ts values where t is → ts values where ts is
In Figure 2 and Figure 5, please write the names of each materials like Figure 1.
In Figure 3, please write the almost scale (size).
In Table2 explanation, please write the no abbreviation word of “SAC” and “FAC”.
Please explain “Pebble Ports Open”.
In Figure 4 photo, please write the scale.
In Table 3, please explain 40,35,25% Fines meanings.
Also 40% Fines →40% Fines (%).
Author Response
Please see the attachment. Each one of the reviewer’s comments were carefully checked and included in the attached reviwed text, as follows (comments in between "" refers to phrases included in the reviwed text, as attached):
- "Cristalino ore deposit (Para – Brazil),"
- "90% passing size (P90) of 19 mm"
- "with an 80% passing size(P80) of 0.21 mm"
- "(Delboni, 1999)"
- "The SABC, an acronym for Semi-Autogenous Ball mill pebble Crusher configuration, "
- "t10 is the percentage passing 1/10th of the initial mean particle size tested"
- The names of each material was included in the referred figures, as shown in the attached file
- "SAG mill (6´x2´), "
- "(*) SAC: semi-autogenous mill with pebble crusher - SAC-A: semi-autogenous mill with pebble crusher in open mode – FAC: fully autogenous test in a closed-circuit configuration"
- "In one test, there were four open pebble ports which consists of relatively large apertures (76 mm x 76 mm) located around the mill grate"
- Please see the sizes in inches of each pile that formed the ore charge
-
"(*) 40% Fines; 35% Fines; 25% Fines – percentage of primary crusher discharge crushed processed on the secondary + tertiary crushing circuit"
- "Table 3 were calculated according to 40%, 35%, and 25% of fines, in this case referring to the amount (%) of tertiary crusher product, which indicated P80 of 76 mm, 81 mm, and 90 mm, respectively"

Reviewer 3 Report
An interesting paper describing the increase in mining productivity based on the use of modern scientific approaches. The case study is clearly presented and provides a complete overview of the whole process.
Author Response
Please see the attachment.
Even though there were no detailed recommendations specific points to review, a thorough review was carried out in the text, thus resulting in the attached file.
